# Evaluation of Water Quality of Collected Rainwater in the Northeastern Loess Plateau

Pengfei Zhang [1,2,*], Menglin Xiao [1], Yanyan Dai [1,2], Zhaorui Zhang [1], Geng Liu [1,2] and Jingbo Zhao [3,4]

[1]  School of Geography Science, Taiyuan Normal University, 319 Daxue Avenue Yuci District, Jinzhong 030619, China
[2]  Institute for Carbon Neutrality, Taiyuan Normal University, 319 Daxue Avenue Yuci District, Jinzhong 030619, China
[3]  School of Geography and Tourism, Shaanxi Normal University, Xi'an 710119, China
[4]  Key Laboratory of Aerosol Chemistry and Physics, Institute of Earth Environment, Chinese Academy of Sciences, Xi'an 710061, China
[*]  Correspondence: zhangpf66@163.com

**Abstract:** Water resources are scarce in the Northeastern Loess Plateau, and water cellar water (WCW) is a vital water resource available in the vast rural areas of the region. The quality of WCW was assessed by principal component analysis (PCA) and Nemerow's pollution index (NPI) for different rainfall catchment areas, depths, and storage times. Eleven indicators were measured, including pH, electrical conductivity (EC), $F^-$, $Cl^-$, $NO_3^-$, $SO_4^{2-}$, $Na^+$, $NH_4^+$, $Ca^{2+}$, $Mg^{2+}$, and $K^+$. The results show that the tap water quality in the rural areas of the Northeastern Loess Plateau is above the second level and meets the drinking water standard (DWS), which is similar to the tap water quality in the region. The main component score of water quality from tile roof + cement ground (I) is 0.32, and the Nemero index is 0.41; the principal component score of water quality from cement ground (I) is 0.45, and the Nemero index is 0.29; the principal component score of water quality from trampled land (I) is 0.59, and the Nemero index is 0.44; the principal component score of water quality from tile roof + trampled land (II) is 1.87, and the Nemero index is 1.10. The rainwater harvesting catchment area of tile roof + cement ground (I) ensured the highest water quality, followed by cement ground (I), trampled ground (I), and tile roof + trampled ground (II). The water quality of the catchment area for artificially collected rainwater (roof tile surface, cement ground, etc.) was better than that of the original soil (trampled ground). The highest water quality was found at a storage time of 1 year (I), followed by 2.5 years (I), and 2 months (II). A depth of 4 m (I) contributed to the highest water quality, followed by 2 m (II), 3 m (II), and 1 m (II). Water quality improved with the increasing depth of WCW. The rainfall and WCW in the area were weakly alkaline, and the groundwater was contaminated with $NO_3^-$. PCA's water quality assessment results were similar to the NPI method, indicating that both methods can be used in combination for unconventional water quality assessment.

**Keywords:** Northeastern Loess Plateau; principal component analysis; Nemerow pollution index; water quality; water cellar water

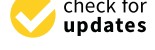

## 1. Introduction

Water scarcity is often a critical limiting factor for industrial development in arid and semi-arid regions [1]. In the Loess Plateau of China, where surface and groundwater resources are often either unavailable or too saline, precipitation is the primary water source for socioeconomic and household livelihood development [2]. In addition, annual precipitation in the northeastern part of the Loess Plateau is generally less than 400 mm and is concentrated in July and August, compared with 1500–2000 mm in the northern part of the Loess Plateau. The available water resources per capita are 110 $m^3$, only 15.3% and 3.7% of the national and global average, respectively [3]. This situation is

particularly pronounced in the rural areas of the Loess Plateau, where residential water use and industrial development are entirely dependent on precipitation.

Water cellars have played a significant role as a vital rainwater harvesting system for collecting episodic rainfall in rural communities of the Loess Plateau since ancient times [3]. A water cellar system consists of a water cellar, a catchment area, a sedimentation tank, and an inlet [3]. In addition, water cellars are usually built-in courtyards along roadsides and cropland sides because these areas can easily collect rainwater. The catchment areas are mainly yards, roofs, and roads covered by bricks, concrete, and trampled soil surfaces. The rainwater collected in the water cellar can effectively protect the soil and water, and the water lost in the rainy season can be collected and used in the dry season, so water resources can be used wisely.

As an unconventional water resource, water cellar water (WCW) mainly converts average atmospheric precipitation into the surface runoff, which then flows into water cellars for irrigation and drinking by residents. The vapor circulation pattern is atmospheric precipitation-surface runoff-water cellar-available water. Poor disinfection facilities, improper manual management, and water cellar materials seriously affect the quality of WCW. In addition, residents in rural areas who have been using WCW for a long time do not care about water quality and sanitation. Most residents use WCW directly as drinking water, and their concern about water quality is limited to judging water taste by experience rather than standards. Due to the unique circulation process of WCW, the quality of WCW has strong regional characteristics. In addition, implementing the rural revitalization strategy has changed the situation in rural areas and many parts of the WCW cycle process (atmosphere, catchment area, water cellar substances). Therefore, the current status of WCW quality in rural areas is of great importance for the safety of the drinking and irrigation water for rural residents.

Since 2013, China has renovated 17.94 million units of dilapidated rural houses (Ministry of Housing and Urban-Rural Development, People's Republic of China, https://www.mohurd.gov.cn/xinwen/gzdt/202003/20200330_244697.html, accessed on 30 March 2020). Since 2016, China has built and renovated 1.388 million km of rural roads, and by the end of 2019, the total length of China's rural roads will be 4.2 million km (Ministry of Transport of the People's Republic of China, https://www.mot.gov.cn/jiaotongyaowen/202010/t20201023_3479335.html, accessed on 23 October 2020). From 2013 to 2019, the urban population increased from 53.73% to 60.6% of the total population, while the rural population decreased from 46.27% to 39.4% (Central Government of the People's Republic of China, http://www.gov.cn/shuju/hgjjyxqk/xiangqing/np.html, accessed on 20 December 2021). Moreover, the rural population is continuing on a decreasing trend. According to the above explanation, on the one hand, with the development of rural facilities and the environment, the catchment area (the main component of water cellars) has improved considerably, and the efficiency of rainwater harvesting has increased significantly. On the other hand, urbanization has reduced the rural population, and this has led to some rural infrastructures, such as water cellars, being left unused.

The total amount of $Ca^{2+}$, $Mg^{2+}$, $K^+$, $Na^+$, $Cl^-$, $NO_3^-$, $SO_4^{2-}$, and $CO_3^{2-}$ ions is about 95% to 99% of the total solutes in natural water. These ions are essential indicators for water quality monitoring and are the main ions used to study the chemical properties of water [4]. Common anions in water ($Cl^-$, $NO_3^-$, $SO_4^{2-}$) and common cations ($Na^+$, $Mg^{2+}$, $Ca^{2+}$, $K^+$) form hardness and alkalinity, and the variation in the amount between anions and cations affects the pH of the water, thus determining the water properties. Ullah et al. [5] conducted a water chemistry analysis in Pakistan for pH, $HCO_3^-$, $Ca^{2+}$, $Mg^{2+}$, $Na^+$, $NO_3^-$, $K^+$, $Cl^-$, and $F^-$; using WQI water quality indicators, 44% of the water in the area was found to be poor-quality water. Khan [6] collected water samples from 43 sites in the Tehir Isahel, Mianwali, Punjab, Pakistan area and measured physical and chemical parameters such as pH, conductivity, total hardness, $Ca^{2+}$, $Mg^{2+}$, $Cl^-$, and $F^-$ concentrations; most of the water was not usable for drinking. There are several methods of water quality evaluation, each with advantages and disadvantages [7]. WCW is an unconventional water

resource which is different from conventional water resources, such as rivers and lakes, so it is important to choose a suitable evaluation method. Singh et al. [8] used principal component analysis to evaluate the water quality of lakes for pH, conductivity, $Ca^{2+}$, $K^+$, $Na^+$, $Cl^-$, $NO_3{}^-$, $SO_4{}^{2-}$, $CO_3{}^{2-}$ and other ions, with good results. Ajani et al. [9] used the Nemerow pollution index to evaluate the integrated pollution status of sediments in the water zone. Principal component analysis (PCA) can avoid subjective arbitrariness and objectively determine the weights of each index for quantitative studies. However, in water quality evaluation, PCA can reduce the influence of water quality indicators that severely exceed the standard [10]. Nemerow's pollution index (NPI) method is one of the water quality index methods that uses statistical values to evaluate the pollution status of water bodies to determine the degree of pollution, qualitatively reflecting the pollution status of water bodies. The evaluation results are intuitive, and the NPI method highlights the impact of evaluation indicators that significantly exceed the standard [11]. At the same time, some studies have concluded that a single water quality evaluation method cannot fully and objectively reflect the water function, showing certain limitations of water quality evaluation. An effective combination of water quality evaluation methods can make the evaluation results more realistic and reliable [12]. In conclusion, the effective combination of PCA and NPI methods can improve the accuracy of the evaluation results and avoid the limitations of a single method.

Given this, in this study, pH, EC, and ions ($F^-$, $Cl^-$, $NO_3{}^-$, $SO_4{}^{2-}$, $Na^+$, $NH_4{}^+$, $Ca^{2+}$, $Mg^{2+}$, and $K^+$) were used as evaluation indexes for WCW quality in different catchments using PCA and NPI methods. The WCW quality of different catchments, depths, and storage times in rural areas of Northwestern Shanxi Province were used as evaluation indicators to provide a scientific basis for ensuring drinking water safety and the rational use of water resources in rural areas.

## 2. Materials and Methods

### 2.1. Study Area

The Northwestern Shanxi province is located in the northeastern part of the Loess Plateau [13]. Northwestern Shanxi is located between longitudes 111°19′ E and 112°52′ E and between latitudes 38°39′ N and 40°17′ N. It covers an area of about $1.42 \times 10^4$ km$^2$ and includes nine counties (Zuoyun, Youyu, Pinglu, Shuocheng, Hequ, Pianguan, Baode, Shenchi, and Wuzhai) (Figure 1) [14]. The region is transitioning from a semi-humid to a semi-arid climate and from shrub-steppe to meadow vegetation [13]. The average annual rainfall in the region is low (400–500 mm, mainly concentrated in June–September). The average annual evaporation varies between 1780 mm and 1950 mm. The average annual temperature ranges between 3.6 °C and 7.5 °C, with monthly average temperatures ranging from −16.0 to −10.0 °C in January and 19.0 to 22.5 °C in July. Winters are dry and cold, while spring is dry and windy. The temperature difference between summer and winter is enormous [14]. The main vegetation types are secondary scrub and grasses.

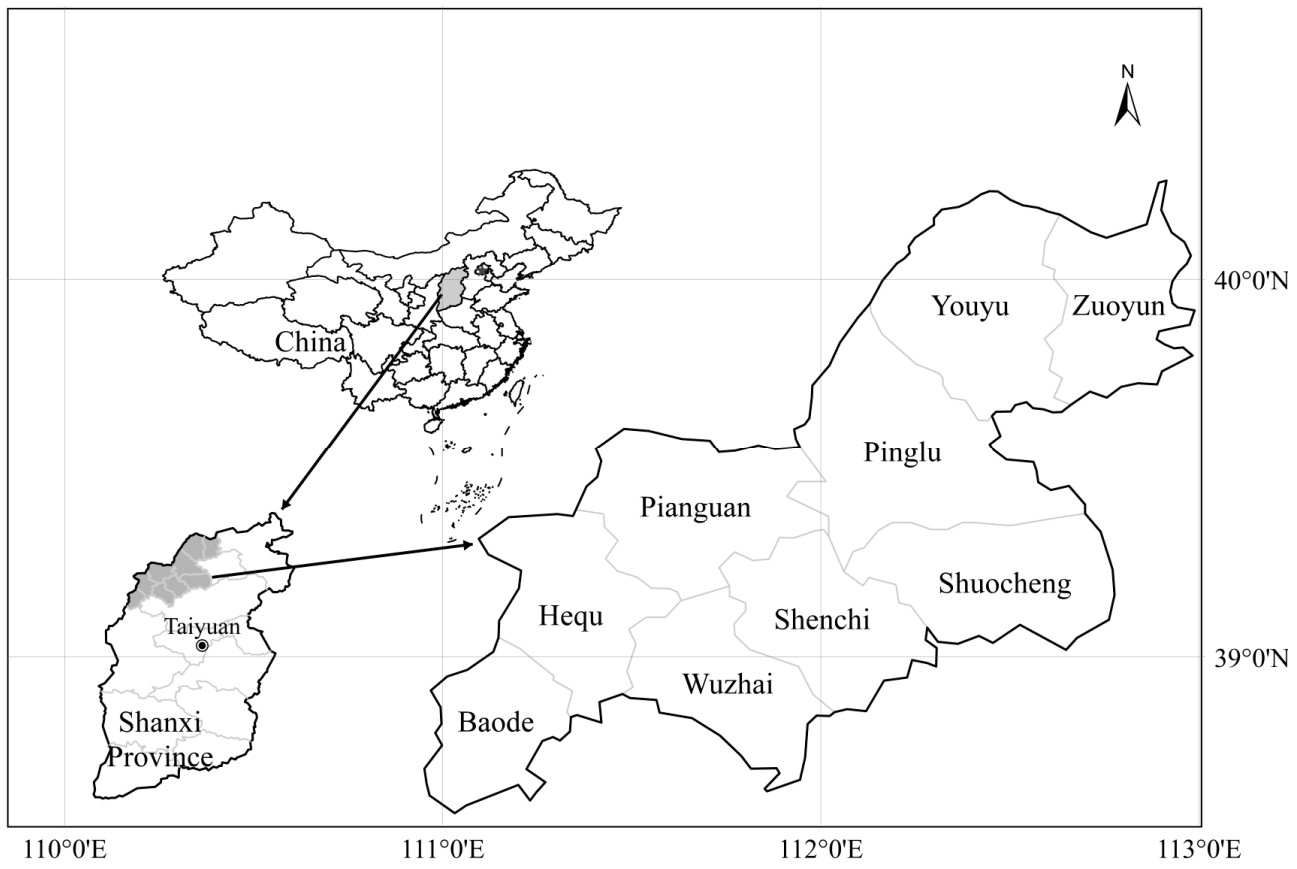

**Figure 1.** Map of study site (Northwest Shanxi province, China).

As of the end of 2019, the region's total population was 1.67 million, of which 940,000 were urban and 730,000 were rural.

*2.2. Methods*

In October 2019, after the rainy season in the study area, six representative dry wells (Nos. 1–6) with similar volumes were selected for sampling, based on field investigation. The characteristics of the dry wells in terms of the catchment area, storage time, and sampling depth are shown in Table 1. Among them, the catchment areas Nos. 1–4 were tile roof + trampled ground (A), cement floor (B), tile roof + courtyard cement floor (C), and trampled ground (D), respectively. The water stored in the cellar is current rainwater and is completely stored at the end of the rainy season. The storage time was about 2 months, and the sampling depth was 2 m above the water surface for analyzing the WCW quality in different catchments. The water cellars Nos. 1, Nos. 5, and Nos. 6 were the same catchment (tile roof + trampled ground), and the storage time was 2 months (T1), 1 year (T2), and 2.5 years (T3), respectively. WCW stored exclusively at the time of sampling was used to analyze the quality of WCW at different storage times. Water samples were collected at 1 m (S1), 2 m (S2), 3 m (S3), and 4 m (S4) from the water surface in water cellar Nos. 1 to analyze the quality of WCW at different depths.

A one-liter portable deep water sampler was used to collect the samples. After sampling, a quarter of the surface layer was poured out, and then the pH and EC values of the water samples were tested and recorded on-site using a portable pH and conductivity meter (CT-6322, Kodak, Shenzhen). The water samples were then filled into cleaned plastic bottles (0.5 L). The samples were sealed, placed in a new box with an ice pack, and quickly returned to the laboratory for analysis and determination. At the same time, rainwater samples were collected from the township, and groundwater and tap water samples from the county, about 20 km away in a straight line. Three parallel samples were collected from

each sampling site. As the samples from 2 m depth in the No. 1 water cellar were used to study the water quality from different catchment areas, storage times, and depths, a total of 33 samples were collected throughout the process. The samples were returned to the laboratory, and the concentration of each ion ($F^-$, $Cl^-$, $NO_3^-$, $SO_4^{2-}$, $Na^+$, $NH_4^+$, $Ca^{2+}$, $Mg^{2+}$, $K^+$) was measured using an ICS-1500 ion chromatography system.

**Table 1.** WCW sampling points in Northwest Shanxi province (11 monitoring points; the sampling was repeated 3 times, for a total of 33 samples).

| Water Cellar Number | Characteristic of Catchment Area | Abbreviation | Storage Time | | Sampling Depth | | Number of Samples |
|---|---|---|---|---|---|---|---|
| | | | Time/Month | Abbreviation | Depth/m | Abbreviation | |
| 1 | Tile roof + trampled ground | A | 2 | T1 | 1 | S1 | 3 |
| | | | | | 2 | S2 | 3 |
| | | | | | 3 | S3 | 3 |
| | | | | | 4 | S4 | 3 |
| 2 | Cement floor | B | 2 | - | 2 | - | 3 |
| 3 | Tile roof + courtyard cement floor | C | 2 | - | 2 | - | 3 |
| 4 | Trampled ground | D | 2 | - | 2 | - | 3 |
| 5 | Tile roof + trampled ground | - | 12 | T2 | 2 | - | 3 |
| 6 | Tile roof + trampled ground | - | 30 | T3 | 2 | - | - |
| 7 | Rainwater | RW | - | - | - | - | 3 |
| 8 | Groundwater | GW | - | - | - | - | 3 |
| 9 | Tap water | TW | - | - | - | - | 3 |

*2.3. Water Quality Evaluation Methods*

2.3.1. PCA

PCA is mainly used to establish a comprehensive index system for WCW quality evaluation and to classify water quality samples. The basic steps [15,16] are as follows.

(1) From the original data $x_{ij}$, the normalization matrix X is obtained, and the correlation coefficient matrix R is calculated. The eigenvalues $\lambda$ and eigenvectors u are obtained, and the number of principal components is determined according to the cumulative contribution rate Q of greater than 80%.

$$a_{ij} = \lambda_{ij} / \sum_{k=1}^{p} \lambda_k \tag{1}$$

$$Q = \sum_{j=1}^{m} a_{ij} \tag{2}$$

In this case, *p* represents the number of eigenvalues, *m* represents the number of principal components, and $\lambda$ represents the eigenvalue.

(2) Based on the principal component scores and the corresponding feature vectors, the principal component of each sample $z_j$ and the total evaluation score $F_j$ are obtained, and the comprehensive evaluation function F is determined. The larger the score, the more serious the pollution is.

$$F_j = X \times z_j (j = 1, 2, \cdots, m) \tag{3}$$

$$F = \sum_{j=1}^{m} (a_j \times F_j) \tag{4}$$

2.3.2. NPI Method

The NPI method was used to determine the degree of water contamination [17]. This paper adopted the "Sanitation Standard for Drinking Water (GB5749-2022)" and selected six parameters: pH, EC, fluoride, chloride, ammonium, and nitrate as evaluation indicators. The calculation steps are as follows:

(1) Sub-index $P_i$:

$$P_i = C_i / L_i \tag{5}$$

In the formula: $C_i$ is the measured concentration for a single indicator, and $L_i$ is the hygiene standard for a single indicator.

(2) pH, EC calculations.

When $C_i > \overline{L_i}$,

$$P_i = \left(C_i - \overline{L_i}\right) / \left(L_{max} - \overline{L_i}\right) \tag{6}$$

When $C_i < \overline{L_i}$,

$$P_i = \left(\overline{L_i} - C_i\right) / \left(L_{max} - \overline{L_i}\right) \tag{7}$$

(3) Nemerow pollution index (*PI*):

$$PI = \sqrt{\left[\left(\sum P_i / n\right)^2 + P_{imax}^{\ 2}\right] / 2} \tag{8}$$

Substances hazardous to health index (*KI*):

$$KI = P_i / P_{it} \tag{9}$$

In the formula: *n* is the number of water quality indicators involved in the evaluation, $P_{it}$ is the risk ratio, and fluoride is a toxicological indicator. If *KI* < 1, there is no health hazard; if $KI \geq 1$, WCW cannot be used as drinking water. The classification of pollution evaluation indicators is shown in Table 2.

**Table 2.** Water quality classification.

| Grade | Pollution Index | Level | Pollution | Grade | Pollution Index | Level | Pollution |
|-------|-----------------|-------|-----------|-------|-----------------|-------|-----------|
| 1 | PI < 1 | I | Clean | 4 | $3 \leq PI < 5$ | IV | Heavy pollution |
| 2 | $1 \leq PI < 2$ | II | Light pollution | 5 | $PI \geq 5$; $KI \geq 1$ | V | Serious pollution |
| 3 | $2 \leq PI < 3$ | III | Moderate pollution | | — | | |

*2.4. Data Analysis*

Excel 2016 was used for data sorting and analyzing, Origin 2019 was used for graphing, and R language 4.0.2 was used for one-way ANOVA analysis of the measured data.

**3. Results**

*3.1. Precipitation Characteristics in Northwestern Shanxi Province*

Based on the statistical data provided by the China Meteorological Administration, three meteorological observation stations, Youyu, Shuocheng, and Kelan in Northwestern Shanxi province, were selected as research subjects. Based on the precipitation data of each station from 1990–2019, the variation in precipitation and evapotranspiration over the years (Figure 2a) and the monthly average precipitation and evapotranspiration in Northwestern Shanxi province (Figure 2b) were plotted. As can be seen from Figure 2a, the annual precipitation in the region from 1990–2019 ranged from 300–600 mm, with an average value of 444.35 mm, and the evaporation ranged from 800–1910 mm, with an average value of 1374.85 mm. In general, the annual precipitation is low, with high interannual variability; the dispersion coefficient (ratio of standard deviation to mean) is 0.15, and the annual evaporation is more than three times the annual precipitation. At the same time, precipitation in the region is highly variable throughout the year, mainly concentrated from June to September. The precipitation in these four months accounts for more than 70% of the total annual precipitation.

In summary, the precipitation in Northwestern Shanxi province can be summarized as poor and erratic, leading to water shortages in the region. The low precipitation and mismatched timing of crop water needs have led to nine droughts in ten years in the region, which is highly detrimental to agricultural production and increases the economic risk for farmers. Therefore, the construction of water cellars and the rational use of WCW are fundamental.

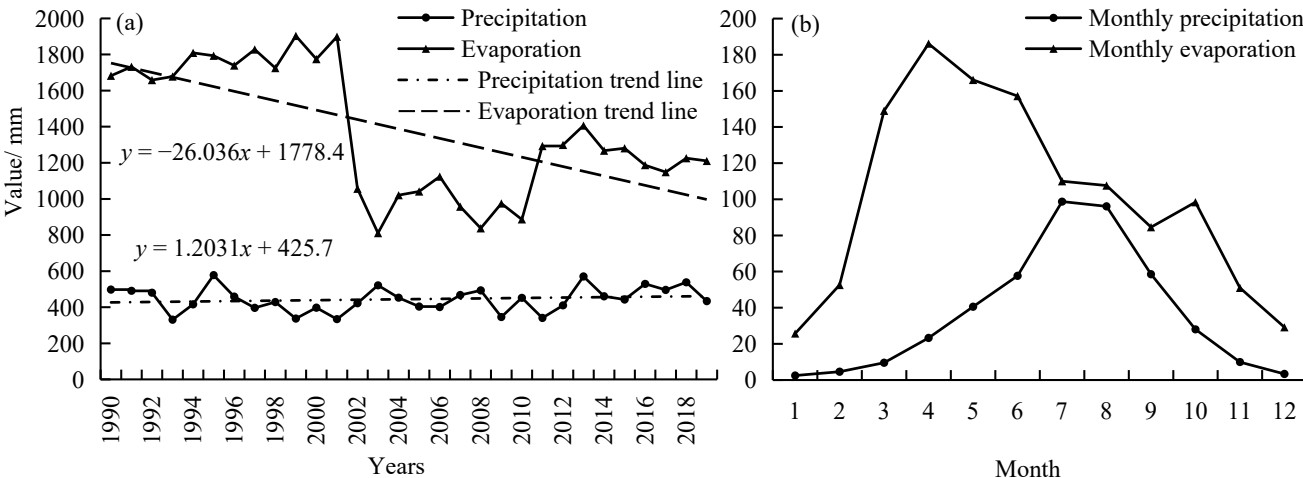

**Figure 2.** Annual precipitation, evaporation (**a**), and monthly precipitation, evaporation (**b**) in Northwest Shanxi province.

### 3.2. Characteristics of Various Types of Water in Rural Areas of Northwestern Shanxi Province

The water quality parameters of WCW in Northwestern Shanxi province were obtained by measurement (Table 3). The results showed that the ion concentrations were $NO_3^- > SO_4^{2-} > NH_4^+ > Ca^{2+} > Mg^{2+} > K^+ > Na^+ > Cl^- > F^-$ in descending order, and the total average concentration of anions (4.92 mg/L) was higher than the total average concentration of cations (1.65 mg/L). The total average concentrations of $F^-$, $Cl^-$, $SO_4^{2-}$, $NO_3^-$, $Na^+$, $NH_4^+$, $K^+$, $Mg^{2+}$ and $Ca^{2+}$ were 0.007 mg/L, 0.07 mg/L, 2.84 mg/L, 2.00 mg/L, 0.11 mg/L, 0.71 mg/L, 0.16 mg/L, 0.17 mg/L, and 0.50 mg/L, respectively. The total concentration of the above nine ions in the WCW was 6.56 mg/L. The average pH and EC values of the WCW were 8.53 and 2.02 ms/cm. In for these monitoring indicators, except for the pH value, which is slightly higher than the drinking water standard (DWS). All other indicators are lower than the DWS. Comparing the water quality indicators at each sampling site, this paper found that the concentration of anions was higher than that of cations, and the pH values were generally high (Table 4).

**Table 3.** The WCW quality parameters in Northwest Shanxi province (Chinese drinking water standards GB 5749—2022).

| Ions | $F^-$ | $Cl^-$ | $NO_3^-$ | $SO_4^{2-}$ | $Na^+$ | $NH_4^+$ | $K^+$ | $Mg^{2+}$ | $Ca^{2+}$ | pH | EC |
|---|---|---|---|---|---|---|---|---|---|---|---|
| | mg/L | | | | | | | | | | ms/cm |
| Minimum | 0 | 0.04 | 0.91 | 0.81 | 0.05 | 0.55 | 0.12 | 0.11 | 0.42 | 7.88 | 1.54 |
| Max | 0.01 | 0.14 | 5.72 | 3.73 | 0.23 | 1.29 | 0.28 | 0.26 | 0.83 | 9.01 | 3.19 |
| Average | 0.007 | 0.07 | 2.84 | 2.00 | 0.11 | 0.71 | 0.16 | 0.17 | 0.50 | 8.53 | 2.02 |
| Standard deviation | 0.00 | 0.04 | 1.76 | 1.12 | 0.05 | 0.20 | 0.04 | 0.06 | 0.12 | 0.48 | 0.52 |
| Rainwater | 0.018 | 0.06 | 1.69 | 1.32 | 0.01 | 0.86 | 0.19 | 0.4 | 0.30 | 7.73 | 2.23 |
| Groundwater | 0 | 0.67 | 17 | 7.31 | 1.065 | 0.24 | 0.05 | 2.91 | 1.18 | 8.3 | 7.81 |
| Tap water | 0.01 | 0.17 | 0.3 | 0.26 | 0.03 | 0.72 | 0.16 | 0.08 | 0.03 | — | — |
| Drinking water standard | 1 | 250 | 10 | 250 | — | — | — | — | — | 6.5–8.5 | 1–4 |

According to the ANOVA significance analysis, the water quality indicators between different rain collection surfaces (A, B, C, D) and different storage times (T1, T2, T3) showed that F-differences were not significant ($p > 0.05$), and the remaining ion concentrations were significantly different ($p < 0.05$). The pH of the cellar water at different depths (S1, S2, S3, S4) all exceeded the standard, and the differences in $Cl^-$, $NO_3^-$, $SO_4^{2-}$, and EC were not significant ($p > 0.05$); the other indicators showed significant differences ($p < 0.05$). None of the other rainwater indicators exceeded the standard; however, the pH value was 7.73, which was slightly alkaline, but still met the drinking water standard. The $NO_3^-$ concentration in the groundwater was 17 mg/L, exceeding the standard, the pH value was

8.3 and was slightly alkaline, the EC value was 7.81 ms/cm, which seriously exceeded the standard, and the concentrations of other indicators were normal. All indicators of tap water were within the scope of the drinking water safety standards.

**Table 4.** Correlation between water quality composition and well depth.

| Correlation | Depths | $F^-$ | $Cl^-$ | $NO_3^-$ | $SO_4^{2-}$ | $Na^+$ | $NH_4^+$ | $K^+$ | $Mg^{2+}$ | $Ca^{2+}$ | pH | EC |
|---|---|---|---|---|---|---|---|---|---|---|---|---|
| Depths | 1 | — | — | — | — | — | — | — | — | — | — | — |
| $F^-$ | 0.29 | 1 | — | — | — | — | — | — | — | — | — | — |
| $Cl^-$ | 0.06 | 0.08 | 1 | — | — | — | — | — | — | — | — | — |
| $NO_3^-$ | −0.4 | −0.17 | 0.07 | 1 | — | — | — | — | — | — | — | — |
| $SO_4^{2-}$ | 0.24 | −0.01 | 0.24 | 0.17 | 1 | — | — | — | — | — | — | — |
| $Na^+$ | 0.78 ** | 0.53 | 0.07 | −0.18 | −0.05 | 1 | — | — | — | — | — | — |
| $NH_4^+$ | 0.57 | −0.06 | −0.01 | −0.42 | −0.22 | 0.62 * | 1 | — | — | — | — | — |
| $K^+$ | 0.45 | −0.13 | −0.13 | −0.36 | −0.43 | 0.58 * | 0.85 ** | 1 | — | — | — | — |
| $Mg^{2+}$ | 0.52 | 0.1 | −0.1 | −0.52 | −0.44 | 0.60 * | 0.83 ** | 0.90 ** | 1 | — | — | — |
| $Ca^{2+}$ | 0.73 ** | −0.17 | −0.12 | −0.51 | −0.02 | 0.32 | 0.60 * | 0.63 * | 0.68 * | 1 | — | — |
| pH | −0.76 ** | 0.23 | 0.16 | 0.45 | −0.06 | −0.32 | −0.57 | −0.55 | −0.57 | −0.97 ** | 1 | — |
| EC | 0.49 | 0.23 | 0 | −0.67 * | −0.02 | 0.45 | 0.51 | 0.65 * | 0.66 * | 0.61 * | −0.47 | 1 |

**—at the 0.01 level (two-tailed), the correlation is significant; *—at the 0.05 level (two-tailed), the correlation is significant.

In Northwestern Shanxi province, the total concentrations of 9 ions in rainwater, groundwater, and tap water were 4.84 mg/L, 30.43 mg/L and 1.76 mg/L, respectively. Of the 11 water quality indicators measured, all followed the DWS, except for $NO_3^-$ and EC in groundwater, which was higher than the DWS.

*3.3. Quality of WCW for Different Catchment*

Water quality indicators for different catchments (A, B, C, D) show (Figure 3a,b) that total ion concentrations are highest in the C catchment and lowest in the A catchment. The lowest $F^-$ ion concentrations were found in each watershed, with no significant differences. In the A catchment, the $Cl^-$, $SO_4^{2-}$, $Na^+$, $NH_4^+$, $Ca^{2+}$, $Mg^{2+}$, and $K^+$ ion concentrations were significantly lower than in the other catchments; the highest $NO_3^-$ ion concentration of 2.06 mg/L was found in this catchment, but it only higher than in the B catchment area. The EC value in the A catchment was much lower than in the other catchments. However, the pH was the highest at 9.01. The B catchment had the highest $SO_4^{2-}$ ion concentration of 2.09 mg/L. The concentrations of $SO_4^{2-}$, $Na^+$ and $Mg^{2+}$ in the B catchment were significantly lower than in the C and D catchments, but higher than in the A catchment; $Cl^-$, $NH_4^+$, and $K^+$ ion concentrations showed the maximum value among all catchments. The EC value was 2.25 ms/cm; the pH value was 7.88, showing the lowest value among all catchments. In the C catchment, $NO_3^-$ ion concentration was the highest at 5.47 mg/L; $Cl^-$, $SO_4^{2-}$, $Na^+$, and $Mg^{2+}$ ion concentrations were the highest among all catchments; $NO_3^-$, $NH_4^+$, $Ca^{2+}$, and $K^+$ ion concentrations were lower than in other catchments. The EC value was 2.45 ms/cm and showed the maximum value among all catchments. The pH value was 7.90. In the D catchment, the $NO_3^-$ ion concentration was the highest at 5.72 mg/L; $NO_3^-$ and $Ca^{2+}$ ion concentrations showed the highest values among all catchments, while the other ion concentrations were moderate among all catchments.

PCA yielded the water quality scores (F value) of different catchments. Catchment A has a comprehensive water pollution score of 1.87 (Figure 4a), the worst among the catchments. At the same time, catchment D has a comprehensive water pollution score of 0.59, which is relatively poor. Catchment B has a combined water pollution score of 0.45, which is good. Catchment C has a combined water pollution score of 0.32, which is better than the other catchments, showing that the water quality is much better than for the other catchments. The NPI method calculated the WCW pollution indexes for different catchments. As can be seen in Figure 5, the PI for catchment A is 1.10, and the KI is 0.006; the PI for catchment B is 0.29, and the KI is 0.015; the PI for catchment C is 0.41, and the KI is 0.012; and the PI for catchment D is 0.44, and the KI is 0.007. The toxicological indicators are consistent with the DWS. The

WCW quality in catchments C and D is class I, with clean water; the quality of the WCW quality in catchment A is class II, with slightly polluted water.

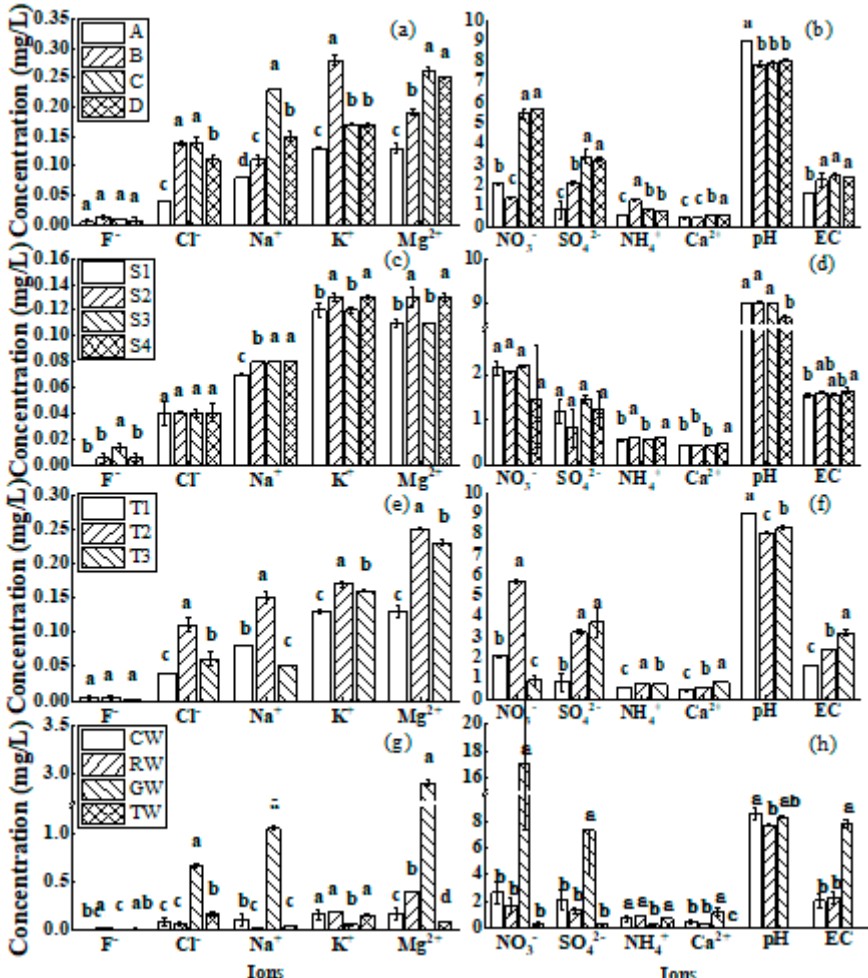

**Figure 3.** Statistical description of the quality index of WCW at different catchment surfaces (**a**,**b**), at different depths (**c**,**d**), at different time (**e**,**f**), and at different water types (**g**,**h**) in Northwest Shanxi province (pH (no unit), EC (ms/cm)). The letters a, b, c, and d in the figure indicate the significant difference (Duncan's test) at the $p = 0.05$ level. Different lowercase letters (a–d) above the bars indicate significant differences (Duncan's test, $p < 0.05$) among the treatments.

In summary, according to the PCA method, WCW quality for different data streams listed from best to worst is C > B > D > A.

According to the NPI method, it can be concluded that the WCW from catchments B, C, and D are clean (grade I), and the WCW from catchment A is lightly polluted (grade II). The results of the WCW quality evaluation in different catchments prove that the results obtained with both the PCA and NPI methods are similar.

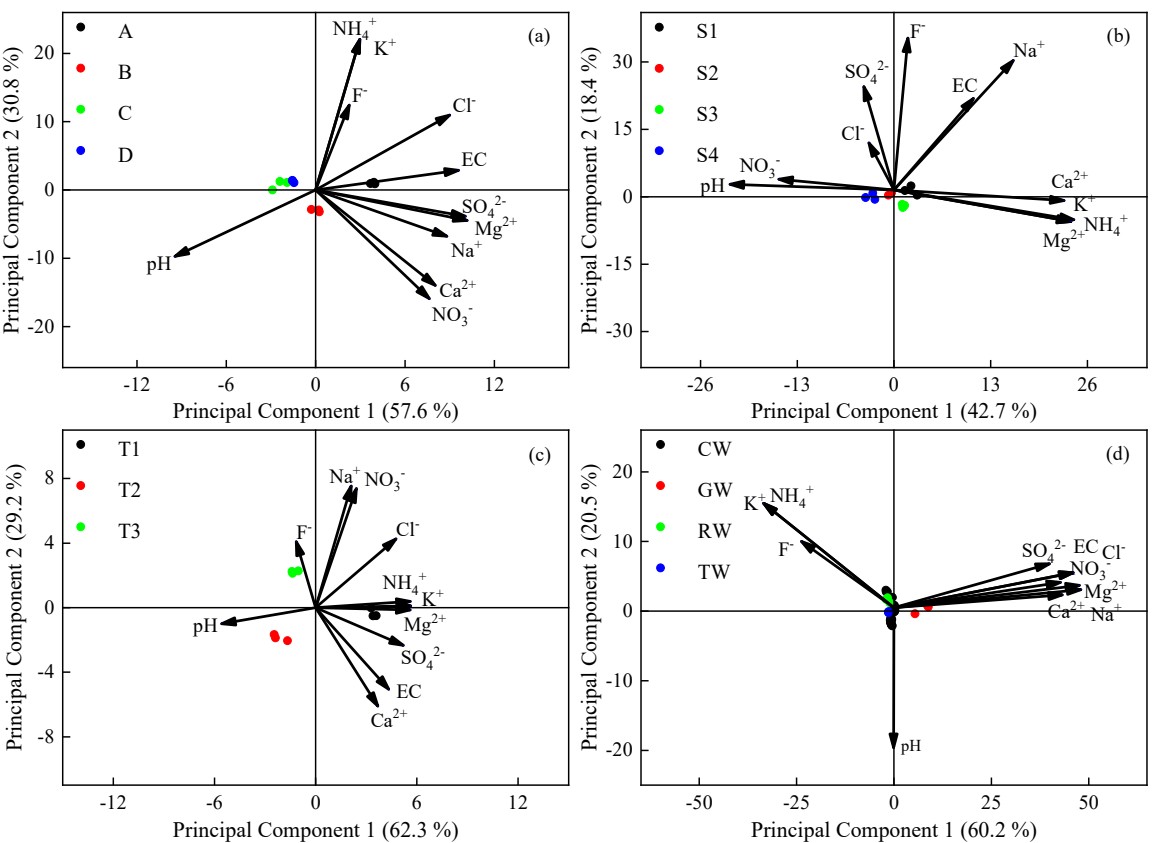

**Figure 4.** Assessment of PCA at different catchment surfaces (**a**), at different depths (**b**), at different storage times (**c**), and with different water types (**d**) in Northwest Shanxi province.

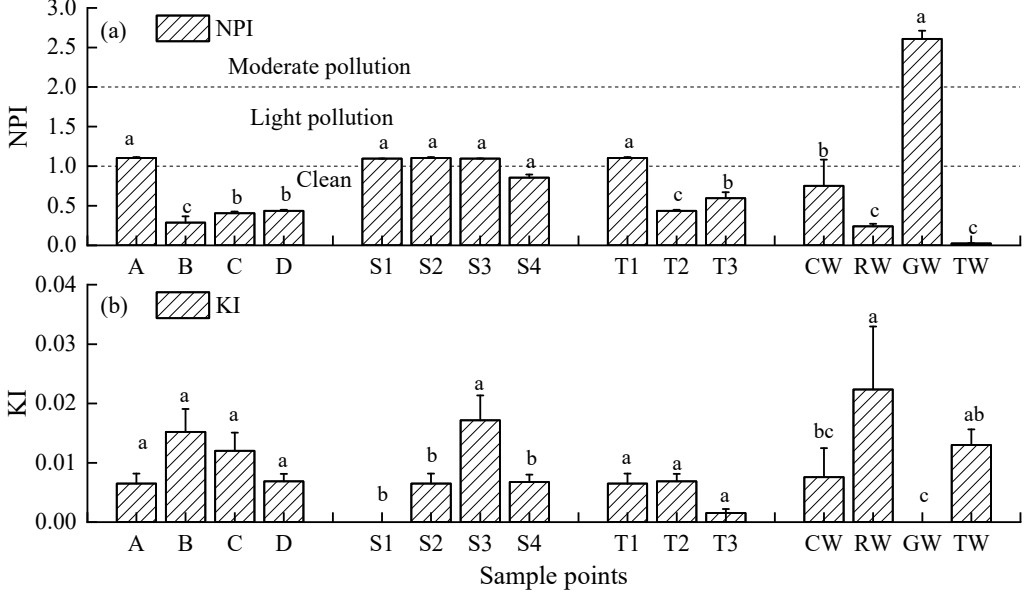

**Figure 5.** Assessment of NPI (**a**) and KI (**b**) of WCW at different catchment surfaces (A, B, C, D), at different depths (S1, S2, S3, S4), at different times (T1, T2, T3), and at different water types (CW, RW, GW, TW) in Northwest Shanxi province (The letters a, b, c, and d in the figure indicate the significant difference (Duncan's test) at the *p* = 0.05 level. Different lowercase letters (a–c) above the bars indicate significant differences (Duncan's test, *p* < 0.05) among the treatments.

### 3.4. Quality of WCW at Different Depths

The water quality indicators at different depths (S1, S2, S3, S4) showed that the ion concentrations were ranked from the highest to the lowest as S3 > S1 > S2 > S4 (Figure 3c,d). At each depth, there were no significant differences in the ion concentrations of $Cl^-$, $NO_3^-$, and $SO_4^{2-}$. Although there were differences in other ion concentrations at each depth, the numerical differences were insignificant ($p > 0.05$, significance analysis). The pH at each depth exceeded the DWS, so the water was alkaline. Although there were differences in EC values at each depth, the numbers did not differ significantly.

The water quality score (F value) of WCW at different depths was obtained by PCA. The total water quality score of S4 was 0.09 (Figure 4b), and the water quality was significantly better than at other depths; the water quality composite score of S2 was 0.5, and the water quality was excellent. Moreover, the water quality composite score of S3 was 0.62, and the water quality was poor; the water quality composite score of S1 was 1.15, and the water quality was the worst. The NPI method was used to calculate the water quality index of WCW at different depths. Figure 5 shows that the PI of S1 was 1.10 and KI was 0; the PI of S2 was 1.10 and KI was 0.007; the PI of S3 was 1.10, and KI was 0.017; the PI of S4 was 0.86 and KI was 0.007. The toxicological indexes all meet the requirements of DWS. The water quality of S4 depth belonged to class I, and the water quality was clean. The water quality of S1, S2, and S3 depths all belonged to class II, and the water quality was mildly polluted. Correlation analysis (Table 4): the depth of the water cellar and the measured water quality components of $Na^+$, $Ca^{2+}$ and pH have a strong correlation (at the 0.01 level, the correlation is significant). $Na^+$ and $Ca^{2+}$ are positively correlated with depth, indicating that with the increase in depth, the ion concentration increases; pH is negatively correlated with depth, indicating that with the increase in depth, the pH value decreases.

In summary, according to PCA, the water quality of WCW at different depths was ranked in descending order: S4 > S2 > S3 > S1. According to the NPI method, the WCW of S4 depth was clean (level I), and the WCW of S1, S2, and S3 depth were lightly polluted (level II). The results of the water quality evaluation of WCW at different depths showed that the results obtained by both the PCA and NPI methods are the same.

### 3.5. Quality of WCW at Different Storage Times

The ion concentrations of WCW for different storage times (T1, T2, T3) increased and then decreased. EC values increased with time, but all met the DWS requirements; pH values decreased and then increased (Figure 3e,f), and the pH value of T1 WCW was 9.01, which exceeded the standard pH value for drinking water. The $F^-$ ion concentration of each storage time was the lowest, and there was no significant difference; $Cl^-$, $NO_3^-$, $Na^+$, $NH_4^+$, $Mg^{2+}$, and $K^+$ first increased significantly and then decreased with the extension of storage time; $SO_4^{2-}$ and $Ca^{2+}$ ions increased significantly with the extension of storage time.

The water quality scores (F values) of WCW with different storage times were obtained by PCA. The water quality of different storage times show significant differences. The combined score of T1 water pollution is 1.76 (Figure 4c), which is the worst water quality; the total score of T3 water pollution is 0.77, indicating good water quality; and the combined score of T2 water pollution is −0.01, showing the best water quality. The NPI method was used to calculate the pollution status of WCW under different storage times. As can be seen from Figure 5, the PI of T1 is 1.10, and the KI is 0.006; the PI of T2 is 0.44, and the KI is 0.007; the PI of T3 is 0.60, and the KI is 0.002. The toxicological indexes all meet the requirements for DWS. The water quality of the T1 water cellar belongs to Class II, and the water quality is lightly polluted; the T2 and T3 water cellars belong to Class I, and the water quality is clean.

In summary, according to the PCA, the quality of WCW under different storage times is in descending order of T2 > T3 > T1. According to the NPI method, WCW with storage times of T2 and T3 are clean (Class I), while WCW with T1 is lightly contaminated (Class II). That is, the quality of T2 (Class I) is better than T3 (Class I), while T3 is better than T1 (Class II). The water quality evaluation of WCW under different storage times showed that the results obtained by both the PCA and NPI methods were the same.

*3.6. Quality of Various Types of Water in Northwestern Shanxi Province*

WCW (24 samples in total), rainwater, groundwater, and tap water were compared at the surveyed sites (Figure 3g,h). The total ion concentrations from highest to lowest were GW (groundwater) > CW (cellar water) > RW (rainwater) > TW (tap water). The ion concentrations of $Cl^-$, $NO_3^-$, $SO_4^{2-}$, $Na^+$, $Ca^{2+}$, and $Mg^{2+}$ in GW water were significantly higher than in other types of water, and the ion concentration of $NO_3^-$ exceeded the DWS; the concentrations of $K^+$ and $NH_4^+$ were significantly lower than for other types of water. A pH of 8.3 and an EC of 7.81 ms/cm seriously exceeded the DWS. The ion concentrations, pH, and EC contents of other types of water were similar and met the DWS requirements.

The PCA obtained the quality score (F value) of each water type. The total score for rainwater was −1.42 (Figure 4d), which is the best quality. Moreover, the total score for tap water was −1.27, which is better; in contrast, the composite score of WCW was −0.53, which is worse. Further, the total score for groundwater was 6.96, which is the worst quality, and is different from other types of water. The NPI method was used to calculate the water quality index of different types of water. As seen in Figure 5, the PI of WCW is 0.75, and the KI is 0.008. The PI of rainwater is 0.24, and the KI is 0.02. The PI of groundwater is 2.61, and the KI is 0. The PI of tap water is 0.02, and the KI is 0.01. The toxicological indexes all meet the requirements of DWS. The quality of WCW, rainwater, and tap water all belong to class I, and the water is clean; groundwater belongs to Class III, which is contaminated.

In summary, the quality of the different types of water were ranked from lowest to highest according to PCA: RW > TW > CW > GW. According to the NPI method, rainwater, tap water, and WCW are all grade I (clean water), while groundwater is grade III (moderately polluted water). Comprehensive analysis shows that rainwater (grade I) is better than tap water (grade I), WCW (grade I), and groundwater (grade III). The quality evaluation results of different types of water show that the results obtained by the PCA and NPI methods are consistent and complementary to each other.

## 4. Discussion

The semi-arid rural areas in the northeastern Loess Plateau have neither groundwater nor surface water, so the residents use WCW for daily drinking water. Therefore, the quality of WCW is directly related to people's health. In rural areas, people's concern about water quality is limited to taste, and the quality of WCW has not yet attracted enough attention from residents.

*4.1. Attribution Analysis of Different Types of WCW Quality in Rural Areas of the Northeastern Loess Plateau*

The results show that the quality of the WCW in the area meets the requirements of the DWS. However, the overall water quality is alkaline. There are three reasons for this result. First, the precipitation in this area is alkaline. Tests on rainwater found that the concentration of each ion did not exceed the standard, and the pH value was 7.73, which is weakly alkaline. A study by Yao et al. [18] found that more than 60% of the 35 rainfall events in Taiyuan in 2012 had pH values greater than 7.0. Second, the rainwater catchment was contaminated. In rural areas, the rainwater catchment is mainly ground and concrete, and the soil and dust in the north are generally alkaline [19]. The raindrops hit the ground when the rainwater catchment area is the ground. Especially in the Loess Plateau, soil aggregates are easily dispersed and disintegrated, and dissolved organic matter in the surface soil is decomposed, leached out, and washed into the rainwater water cellar. The release of soil ions increases the pH of the WCW and the concentration of pollutants in the WCW [19]. When the stormwater catchment is concrete, fine-grained materials, such as sediment and dust, are scoured into the water cellar by rainwater migration and transformation [20]. Third, a series of physical and chemical reactions occur after rainwater enters the water cellar. Water cellars in the rural areas of the Northeastern Loess Plateau are mainly composed of cement and soil. Ordinary Portland cement is highly alkaline

and precipitates alkaline substances, such as calcium hydroxide, during hydration and hydrolysis, which raises the pH value.

The WCW quality of different rainwater catchment areas from best to worst is tile roof + courtyard cement floor (C) > cement floor (B) > trampled ground (D) > tile roof + trampled ground (A). In general, the quality of WCW from artificially treated stormwater catchments (tile roof, cement floor, and others) is better than that on original soil (trampled ground). This is because it is difficult for pollutants to adhere to the artificially treated rainwater catchment, and the catchment is easy to clean. At the same time, to prevent leakage, most residents in this area put plastic sheeting and other clean waterproofing materials on their tile roofs before the rainy season arrives. When it rains, rainwater quickly collects and flows into the water cellar, reducing contact time with surface pollutants and the atmosphere [21]. However, soil particles in the original soil (trampled ground) are easily adhered to by pollutants, such as livestock manure, fertilizers, and dead branches and leaves. When raindrops hit, the soil particles and pollutants are mixed into the water cellar during the rainwater flushing. In addition, even if there are small amounts of dust and sediment particles on the surface of the manually treated rainwater collection, they will enter the water cellar with the rainwater during the initial stage of water collection. Furthermore, no new pollutants will be generated. However, in the original soil, pollutants continue to form and pool as the raindrops hit and are washed away. Eventually, they flow into the water cellar. This further confirms that the catchment is the leading cause of WCW contamination.

The trend in water quality at different depths is 4 m > 2 m > 3 m > 1 m. Surface water is contaminated because the open intake of the surface water cellar dissolves pollutants from the atmosphere and those attached to the intake bucket. With the deepening of the water level, ions interact with rocks and aquifer materials. According to [22], some contaminants may be adsorbed on the cellar walls, and water quality gradually improves through the interaction of gravity and diffusion [23]. The uptake and release of sediment contaminants are a complex process influenced by many factors, which are constantly changing [24]. Meanwhile, the concentration of sediment (containing calcium compounds) in the WCW increases as the water level deepens. Cellar wall cement (containing sodium compounds and calcium compounds) reacts with water to form alkaline substances, thus inhibiting the migration and transformation of contaminants in the WCW. As a result, contaminants are precipitated and enriched in the sediment. In addition, sediments have a sorption effect on pollutants in the water column, and sediment particles sorb heavy metals through physical and chemical processes [25,26]. In addition, the sediments are rich in microorganisms that consume dissolved oxygen from the bottom water to degrade organic pollution [27], so the quality of WCW at 4 m is the best.

With the increase in storage time, the quality of WCW showed a tendency toward better and then worse quality: 1 year > 2.5 years > 2 months. The poor quality of WCW stored for two months was mainly due to the inflow of pollutants, such as sediment and manure, into the water cellar during the rainwater confluence. Due to the short storage time, suspended particles in the water cellar were not sufficiently adsorbed and precipitated. In addition, the hydration and hydrolysis of ordinary Portland cement on the cellar wall produced alkaline substances, such as calcium hydroxide, which resulted in a high pH value at the beginning of storage [23], so the quality of WCW at the beginning of the collection was poor. As microorganisms decompose organic matter to produce carbon dioxide and form carbonic acid to neutralize the alkaline materials in WCW, the pH gradually decreases and tends to be flat [28]. In addition, pH value is a significant factor affecting the adsorption and release of heavy metals in sediments [29,30]. The water environment in the WCW is alkaline, suggesting that high concentrations of $OH^-$ ions can neutralize the positive charge, thus allowing the WCW substrate to adsorb heavy metals, particles, and other contaminants and reduce the concentration of contamination in the WCW [21]. However, since WCW is stagnant water, during too long a storage period (2.5 years), many bacteria and microorganisms can appear in the water, leading to the deterioration of WCW [28].

*4.2. Attribution Analysis of the Quality Differences of Various Types of Water in the Northeastern Loess Plateau*

The quality of different types of water in the Northeastern Loess Plateau was ranked from best to worst as RW > TW > CW > GW. A total of 54.55% of WCW in the study area was clean, and 45.45% was lightly polluted, which met the DWS requirements. The drinking water of rural residents in the northeastern part of the Loess Plateau is mainly WCW. The study shows that the WCW in the area is suitable for drinking, with scientific management.

The data show that $NO_3^-$, $SO_4^{2-}$, $NH_4^+$, $Mg^{2+}$, and $Ca^{2+}$ are the dominant ions of precipitation in the northeastern part of the Loess Plateau, similar to the precipitation composition of other cities in Northern China [18]. The ratio of $SO_4^{2-}/NO_3^-$ concentration is often used to determine the acid-base influencing factors of atmospheric precipitation. A related study found that the $SO_4^{2-}/NO_3^-$ concentration ratios in Taiyuan were 19.02 [31] and 4.38 [18] in 1986 and 2012, respectively. This study's $SO_4^{2-}/NO_3^-$ concentration ratio was 0.78, reflecting that the precipitation in the Northeastern Loess Plateau has changed from sulfation to nitrification. This is related to the forced-flue gas desulfurization from coal-fired emissions in recent years and the increase in $NO_x$ emissions due to the rapid growth of vehicle ownership in China.

The evaluation results showed that the groundwater in the northeastern part of the Loess Plateau was contaminated and was closely related to the Datong and Pingshuo mines located in the northeastern part of the Loess Plateau. Feng et al. [32] studied the groundwater in the Pingshuo mines based on the t-SNE evaluation model and the integrated evaluation method. They concluded that the groundwater quality in the local shallow loose sediments deteriorated, and most of the groundwater was Class III (with some groundwater of Class V). Nan Yalin et al. [33] confirmed that only 29.3% of the groundwater in the Shenmu-Fugu mining area in the Northeastern Loess Plateau was uncontaminated. The leading indicators of groundwater contamination in the Shenmu-Fugu mining area were nitrate and fluoride ions. This is similar to the findings of the present study. In this study, the concentration of $NO_3^-$ ions in groundwater accounted for 55.9% of the total ion concentration, much higher than the concentration of $NO_3^-$ ions in other types of water. In addition, some studies have confirmed that groundwater nitrogen pollution is strongly related to the distribution of mining areas. In dense mining areas, production wastewater, exhaust gases, and wastes containing nitrogen sources contaminate groundwater through infiltration, precipitation, and leaching [32].

The PCA's water quality evaluation results reflect no significant difference between tap water in the northeastern part of the Loess Plateau and WCW in rural areas. Although the quality of tap water obtained by the NPI method is significantly better than that of WCW, the tap water is clean (Class I) after comprehensive analysis. The above findings further demonstrate that the WCW in rural areas of the Northeastern Loess Plateau is clean and can be used for human and animal consumption after scientific management [34,35].

## 5. Conclusions

The following conclusions were drawn from a study of WCW quality in rural areas of the Northeastern Loess Plateau and other types of water resources in the region.

(1) The WCW in rural areas in the northeastern part of the Loess Plateau is clean, less polluted, and of similar quality to tap water, making it suitable for residential use.

(2) Catchment areas significantly impact the quality of WCW. The quality of WCW collected from rainwater catchment areas (such as tile roofs and cement floors) is better than the quality of WCW from the original soil.

(3) The quality of WCW stored in a closed water cellar for about one year was the best, and the longer and shorter the storage time, the worse the quality. In the same water cellar, the quality of WCW at different depths was slightly different, with the surface water having the worst quality and the bottom water having the best quality.

(4) The pH of WCW and rainwater in the study area is generally high, and both WCW and rainwater are weakly alkaline. Groundwater in the Northeastern Loess Plateau is mainly contaminated by $NO_3^-$ ions.

**Author Contributions:** Conceptualization, P.Z., J.Z., G.L. and M.X.; writing—original draft, P.Z., Z.Z., M.X. and Y.D.; writing—review and editing, P.Z., G.L., Y.D. and M.X. All authors have read and agreed to the published version of the manuscript.

**Funding:** This study was supported by the Ministry of Education of Humanities and Social Science project (19YJAZH066), the Shanxi Scholarship Council of China (2020-138), the key research base project of Humanities and Social Sciences in Shanxi Province (20200133), the Scientific and Technological Innovation Programs of Higher Education Institutions in Shanxi (2019L0781 and 2021L436), and the Shanxi Federation of Social Sciences (2020YY207).

**Institutional Review Board Statement:** Not applicable.

**Informed Consent Statement:** Not applicable.

**Data Availability Statement:** All data generated or analyzed during this study are included in the published article.

**Conflicts of Interest:** The authors declare there is no conflict of interest regarding the publication of this paper.

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
