# Peer review of "Evaluation of Water Quality of Collected Rainwater in the Northeastern Loess Plateau"

_sustainability, doi:10.3390/su141710834_

Round 1
Reviewer 1 Report
This work presents a quality assessment of water cellars water (WCW) in the Loess Plateau of China using multiple physio-chemical indexes, further evaluated by principal component analysis (PCA) and Nemerov’s pollution index (NPI). This kind of evaluation process will entail in getting a getting a sense of the water quality meeting the regulatory standards for domestic consumption. It will allow implementation of requisite changes to the infrastructure for better management of water resources in regions facing water challenges.
Introduction- There is a lack of recent literature discussion on implementation of water quality evaluation processes and its associated parameters chosen. What is the rationale for choosing these 11 indexes? It will give the background to your research and explain the chosen parameters/methods to readers.
The reference list is a bit dated, include some recent references after 2020. Kindly cite then and discuss those process investigated in the studies and how the current studies add to the field.
Results and discussion section- There is a lack of proper scientific discussion on the results obtained and the explanation for the effect of investigated index parameters used during the research. Needs to be improved.
Some other points should be addressed in order to complete and clarify some aspects in the manuscript :-
Abstract: Needs improvement and should contain more quantitative data/results rather than being descriptive in nature.
Graphical abstract: I see a repeat of the text in here from the abstract/Highlights section. This section should stand on each own and is better served with graphics/plots illustrations than just repeat text in box.
Line 74: “particularity”, Not the correct placement of the word, check grammar, replace with an appropriate word
Line 87-88: “trend of rural…’odd structure and tense, re-write,
Line 91: “idle” what does it mean? State clearly and elaborate its lack of utilization…
Line 246-247: It was mentioned that concentration of anions was higher”….quantify by how much and what are the implications of each from a water quality perspective
Line 255: Is it total ion concentration?
Line 299: Stating “numerical differences were not large” is a very qualitative range…needs quantification. By how much and is it statistically significant?
Section 3.5/Line 320: Relevance of each parameter pertaining to water quality and its impact must be outlines, either here or in the discussion section
Fig3/Line 371: The Y-axis in the fig needs to have units, what concentration?
Line 363: Give some explanation of what constitutes Grade-I and Grade-III for reference
Line 423: ”catchment no longer” check grammar
Line 427: Expand the reasons for this sequence “4m>2m>3m>1m” not being continuous…What could be the reasons for 2m>3m and not the other way around.
Line 427-439: Improve the discussion with more literature supporting the claims.
Line 452: cite reference
Author Response
First, we are appreciating to you for positive evaluation our manuscript, and we also thanks for giving us a chance to improve our manuscript. All modifications are marked by yellow in revised manuscript, and we commissioned experts in native language to correct the English expressions.
- Question: Introduction- There is a lack of recent literature discussion on implementation of water quality evaluation processes and its associated parameters chosen. What is the rationale for choosing these 11 indexes? It will give the background to your research and explain the chosen parameters/methods to readers.
(Response)
Thanks for your suggestions. We added some references and added reasons to choose these 11 indexes. As shown in the L108-120, L123-126.
The total amount of Ca2+, Mg2+, K+, Na+, Cl-, NO3-, SO42-, and CO32- ions in natural water is about 95% to 99% of the total solutes in natural water. These ions are essential indicators for water quality monitoring and are the main ions used to study the chemical properties of water (Huang et al., 2020). Also, common anions in water Cl-, NO3-, S042- and common cations Na+, Mg2+, Ca2+, K+ form hardness and alkalinity and the variation in the amount between anions and cations affects the pH of the water and thus determines the water properties. Ullah et al. (2022) conducted water chemistry analysis in Pakistan for pH, HCO3- , Ca2+ , Mg2+ , Na+ , NO3- , K+ , Cl- , and F-, using WQI water quality indicators, 44% of the water quality in the area was found to be poor-quality water. Khan (2021) collected water samples from 43 sites in the Tehir Isahel area and measured physical and chemical parameters such as pH, conductivity, total hardness, Ca2+, Mg2+, Cl-, and F- concentrations, most of the water was not usable as drinking water.
Singh et al. (2020) used principal component analysis to evaluate the water quality of lakes for pH, conductivity, Ca2+, K+, Na+, Cl-, NO3-, SO42-, CO32- and other ions with good results. Ajani et al. (2021) used the Nemerow pollution index to evaluate the integrated pollution status of sediments in the water zone.
- Question:The reference list is a bit dated, include some recent references after 2020. Kindly cite then and discuss those process investigated in the studies and how the current studies add to the field.
(Response)
Thanks for your suggestion, we have updated and added some references.
- Question: Results and discussion section- There is a lack of proper scientific discussion on the results obtained and the explanation for the effect of investigated index parameters used during the research. Needs to be improved.
(Response)
Thanks for your advice. We added correlation analysis of ions in L277-289, Section 3.4/ L354-359.
According to the ANOVA significance analysis, the water quality indicators between different rain collecting surfaces (A, B, C, D) and different storage times (T1, T2, T3) showed that F-differences were not significant (P>0.05), and the remaining ion concentrations were significantly different (P<0.05). ). The pH of the cellar water at different depths (S1, S2, S3, S4) all exceeded the standard, and the differences in Cl-, NO3-, SO42-, and EC were not significant (P>0.05), and the other indicators had significant differences (P<0.05). All indicators of rainwater did not exceed the standard, only the pH value was 7.73, which was slightly alkaline, but met the drinking water standard. The NO3- concentration in groundwater was 17 mg/L exceeding the standard, the pH value was 8.3 and was slightly alkaline, and the EC value was 7.81 mS/cm, which was seriously exceeding the standard, and the concentrations of other indicators were normal. All indicators of tap water are within the scope of drinking water safety standards.
Correlation analysis : the depth of the water cellar and the measured water quality components of Na+, Ca2+ and pH have a strong correlation (at the 0.01 level, the correlation is significant). Na+ and Ca2+ are positively correlated with depth, indicating that with the increase in depth, the ion concentration increases; pH is negatively correlated with depth, indicating that with the increase in depth, the pH value decreases.
- Question: Abstract: Needs improvement and should contain more quantitative data/results rather than being descriptive in nature.
(Response)
Thanks for your advice. We revised the abstract and added data. As shown in the L37-42.
- Question: Graphical abstract: I see a repeat of the text in here from the abstract/Highlights section. This section should stand on each own and is better served with graphics/plots illustrations than just repeat text in box.
(Response)
Thanks for your advice. We revised the graphical abstract.
6. Question: Line 74: “particularity”, Not the correct placement of the word, check grammar, replace with an appropriate word.
(Response)
We changed the “particularity of the” to “unique”. As shown in the L85.
7. Question: Line 87-88: “trend of rural…’odd structure and tense, re-write.
(Response)
We changed the sentence “Moreover, the rural population is still on a decreasing trend.”. As shown in the L100-101.
8. Question: Line 91: “idle” what does it mean? State clearly and elaborate its lack of utilization…
(Response)
Water cellars are mostly located in rural areas. Due to the transfer of rural population to cities, the use of water cellars in rural areas is greatly reduced. Therefore, idle means that some water cellars and other rural infrastructure are not used. Meanwhile, we added clear description about “On the other hand, urbanization has reduced the rural population, and some water cellars and other rural infrastructure are not being used. This has led to rural infrastructure such as water cellars being left unused.”. As shown in the L105-107.
9. Question: Line 246-247: It was mentioned that concentration of anions was higher”….quantify by how much and what are the implications of each from a water quality perspective.
(Response)
We revised the sentence “and the total average concentration of anions (4.92 mg/L) was higher than the total average concentration of cations (1.65 mg/L).”
We added the content of anions and cations. As shown in the L266-268.
10. Question: Line 255: Is it total ion concentration?
(Response)
It is total ion concentration, we added “total”. As shown in the L296.
11. Question: Line 299: Stating “numerical differences were not large” is a very qualitative range…needs quantification. By how much and is it statistically significant?
(Response)
significance analysis of ions at different depths, the results show that the significance of the homogeneity of variance test is greater than 0.05, indicating that significant analysis can be performed, and the significance in ANOVA is greater than 0.05 (P>0.05), indicating that Cl-, NO3-, SO42- did not differ significantly at different depths.
We added ANOVA analysis and P-values to the manuscript. As shown in the L339-340.
12. Question: Section 3.5/Line 320: Relevance of each parameter pertaining to water quality and its impact must be outlines, either here or in the discussion section.
(Response)
Thanks for your advice, we added correlation analysis, as shown in the Table 4, L354-359.
Correlation analysis : the depth of the water cellar and the measured water quality components of Na+, Ca2+ and pH have a strong correlation (at the 0.01 level, the correlation is significant). Na+ and Ca2+ are positively correlated with depth, indicating that with the increase in depth, the ion concentration increases; pH is negatively correlated with depth, indicating that with the increase in depth, the pH value decreases.
13. Question: Fig3/Line 371: The Y-axis in the fig needs to have units, what concentration?
(Response)
Thanks for your correction. We added units in Fig.3.
14. Question: Line 363: Give some explanation of what constitutes Grade-I and Grade-III for moderate reference.
(Response)
According to the Nemerow Pollution Index, grade I refers to clean water, and grade III refers to pollution. Rainwater, tap water, and WCW were all grade I water (clean water), while groundwater was grade III water (moderately polluted water). WCW are all grade I (clean water), while groundwater is grade III water (moderately polluted water). As shown in the L414-415.
15. Question: Line 423: “catchment no longer” check grammar.
(Response)
In addition, even if there are small amounts of dust and sediment particles on the surface of the manually treated rainwater collection, they will enter the water cellar with the rainwater during the initial stage of water collection. Furthermore, no new pollutants will be generated.
We revised it. As shown in the L459-463.
16. Question: Line 427: The reasons for this sequence “4m>2m>3m>1m” not being continuous.
(Response)
This sequence is obtained by principal component analysis. According to the Nemerow pollution index, the S2 pollution index is 1.10 and the S3 pollution index is 1.10, both of which are classified as light pollution. As shown in the L466-468.
17. Question: Line 427-439: Improve the discussion with more literature supporting the claims.
(Response)
We added more literature in the manuscript. As shown in theL471, L473-474, L481-482.
“With the deepening of the water level, ions interact with rocks and aquifer materials.” (Loaiza, J.G.; Bustos-T errones, Y .; Bustos-T errones, V.; Monjardín-Armenta, S.A.; Quevedo-Castro, A.; Estrada-V azquez, R.; Rangel-Peraza, J.G. Evaluation of the Hydrochemical and Water Quality Characteristics of an Aquifer Located in an Urbanized Area. Appl. Sci. 2022, 12, 6879. https://doi.org/10.3390/app12146879.)
“The uptake and release of sediment contaminants are a complex process influenced by many factors and constantly changing.” (Wang, T., Xu, S., & Liu, J. (2017). Dynamic Assessment of Comprehensive Water Quality Considering the Release of Sediment Pollution. Water, 9(4), 275. doi:10.3390/w9040275.)
“and sediment particles sorb heavy metals through physical and chemical processes.” (Huang, Z., Liu, C., Zhao, X., Dong, J., & Zheng, B. (2020). Risk assessment of heavy metals in the surface sediment at the drinking water source of the Xiangjiang River in South China. Environmental Sciences Europe, 32(1). doi:10.1186/s12302-020-00305-w. Cabral, J.B.P .; Nogueira, P .F.; Becegato, V .A.; Becegato, V .R.; Paulino, A.T. Environmental Assessment and Toxic Metal-Contamination Level in Surface Sediment of a Water Reservoir in the Brazilian Cerrado. Water 2021, 13, 1044. https://doi.org/10.3390/w13081044.)
18. Question: Line 452: cite reference.
(Response)
We added the reference. As shown in the L497.
Riba, I., García-Luque, E., Blasco, J., & DelValls, T. A. (2003). Bioavailability of heavy metals bound to estuarine sediments as a function of pH and salinity values. Chemical Speciation & Bioavailability, 15(4), 101–114. doi:10.3184/095422903782775163.
Iordache, A.M., Nechita, C., Zgavarogea, R. et al. Accumulation and ecotoxicological risk assessment of heavy metals in surface sediments of the Olt River, Romania. Sci Rep 12, 880 (2022). https://doi.org/10.1038/s41598-022-04865-0.

Reviewer 2 Report
The reviewed work by Zhang et al. addresses an important current problem of household water economy.
The manuscript has several strengths. Highlights include the quality of the collected water cellar water (WCW), a wide range of investigations in terms of the types of WCW water supply and the amount of chemical components determined, a data analysis based on two methods, and a clear workflow and practical conclusions.
However, there are a number of points in the paper that should be corrected in order for the paper to be considered for publication in Sustainability.
Introduction:
- The introduction should include the current state of the art in rainwater harvesting and water quality testing in such reservoirs. It should draw on a wider range of world literature. In general, the weakness of the work is its local character. Of course, research is usually done locally, but the work should be placed in a wider context if the authors' intention is to publish it in a the international journal with global range.
Methods:
- A diagram of the types of WCW tested should be included for a better presentation of the research.
- Information is needed on the number of water samples taken from each rainwater harvesting variant.
Results:
- Figures 3 and 4 should be described in more detail in the caption. Each drawing should be a complete reference object so that the reader does not have to search for explanations in the text (which is tedious). The symbols denoting each component of the figures (a) - h) in Fig. 3 and a) - d) in Fig. 4) should be explained, and symbols a-d should be replaced by others to avoid confusion, and explained under both figures.
Discussion:
- The discussion should be conducted with the results of similar research papers from the world literature. The scope of works cited should be significantly expanded to include results from other countries and continents.
- The work presents a comparison of the determined values of the chemical indicators of water with the standards for drinking water. These standards also include microbiological indicators, which are essential for drinking water. Have such studies been done on water at WCW? Include a comment on this.
In conclusion, the manuscript should be revised and additional review should be undertaken. It is my condition to consider it to be published in Sustainability.
Author Response
Dear Reviewer 2:
First of all, we are grateful to you for approval our manuscript, and we also thanks for your suggestions for improving the manuscript. All modifications are marked by yellow in revised manuscript, and we commissioned experts in native language to correct the English expressions.
- Question: Introduction: The introduction should include the current state of the art in rainwater harvesting and water quality testing in such reservoirs. It should draw on a wider range of world literature. In general, the weakness of the work is its local character. Of course, research is usually done locally, but the work should be placed in a wider context if the authors' intention is to publish it in a the international journal with global range.
(Response)
Thank you very much for your professional comments. We have added some references. As shown in the L114-120, L123-126.
Ullah Z et al. (2022) conducted water chemical analysis on pH, HCO3-, Ca2+, Mg2+, Na+, NO3-, K+, Cl-, F-, etc. in Pakistan, and used the WQI water quality index to find that 44% of the water quality in the area was of poor quality water. Hizbullah Khan et al. (2021) collected water samples from 43 locations in the Tehir Isahel area, and measured physical and chemical parameters such as pH, electrical conductivity, total hardness, Ca2+, Mg2+, Cl-, F- concentrations, most water cannot be used as drinking water.
Singh et al. (2020) conducted a comprehensive evaluation of pH, conductivity, Ca2+, K+, Na+, Cl-, NO3-, SO42-, and CO32- ions of lake water quality using principal component analysis, and obtained good results. Ajani et al. (2021) used the Nemerow pollution index to evaluate the comprehensive pollution status of sediments in the water area.
All new references:
Ajani EG, Popoola OS, Oyatola OO. Evaluation of the pollution status of Lagos coastal waters and sediments, using physicochemical characteristics, contamination factor, Nemerow pollution index, ecological risk and potential ecological risk index. International Journal of Environment and Climate Change, (2021). doi:10.9734/IJECC/2021/V11I330371.
Cabral JBP, Nogueira PF, Becegato VA, Becegato VR, Paulino AT. Environmental Assessment and Toxic Metal-Contamination Level in Surface Sediment of a Water Reservoir in the Brazilian Cerrado. Water (2021) 13, 1044. https://doi.org/10.3390/w13081044.
Huang L, Zhu ZL, Tang XZ, Yuan GF, Zhang XY, Sun XM, Wang J, Chang XX, Cheng YS, Chu GW, Dai GH, Du J, Fu W, Guo SW, Guo YP, He QH, Jiang J, Jiang ZD, Lai JB, Lan ZD, Li M, Li SW, Li W, Lin JH, Lin YB, Liu WJ, Liu XL, Liu XP, Liu YG, Lu ZY, Lu YZ, Su HX, Tang JL, Wang JF, Wang XL, Yang FT, Yin CM, Zhang ZS, Zhao CM, Zhao CY, Zhu MN. A Dataset of eight ions in the water at stations of Chinese ecosystem research network (CERN) during 2004-2016 [J]. China Scientific Data, (2020) 5(02): 110-122.
Huang, Z., Liu, C., Zhao, X., Dong, J., & Zheng, B. Risk assessment of heavy metals in the surface sediment at the drinking water source of the Xiangjiang River in South China. Environment Sciences Europe, (2020) 32(1). doi:10.1186/s12302-020-00305-w
Iordache AM, Nechita C, Zgavarogea R. et al. Accumulation and ecotoxicological risk assessment of heavy metals in surface sediments of the Olt River, Romania. Sci Rep 12, 880 (2022). https://doi.org/10.1038/s41598-022-04865-0.
Khan H. Assessment of Drinking Water Quality of Different Areas in Tehsil Isa Khel, Mianwali, Punjab, Pakistan. Pakistan Journal of Analytical & Environmental Chemistry. (2021), 2221-5255. doi: http://dx.doi.org/10.21743/pjaec/2021.12.16.
Loaiza JG, Bustos-T errones Y, Bustos-T errones V, Monjardín-Armenta SA, Quevedo-Castro A, Estrada-V azquez R, Rangel-Peraza JG. Evaluation of the Hydrochemical and Water Quality Characteristics of an Aquifer Located in an Urbanized Area. (2022) 12, 6879. https://doi.org/10.3390/app12146879.
Riba I, García-Luque E, Blasco J, DelValls TA.. Bioavailability of heavy metals bound to estuarine sediments as a function of pH and salinity values. Chemical Speciation & Bioavailability (2003), 15(4), 101-114. doi:10.3184/095422903782775163.
Singh S K, Harshit S, Harshit J, Narang M. Water quality assessment of a water body using principal component analysis-Sanjay Lake, New Delhi, India. Asian Journal of Water, Environment and Pollution (2020) 17(4). doi:10.3233/AJW200047.
Ullah, Z, Xu, Y, Zeng, XC, Rashid A, Ali A, Iqbal J, Almutairi MH, Aleya L, Abdel-Daim MM, Shah M. Non-Carcinogenic Health Risk Evaluation of Elevated Fluoride in Groundwater and Its Suitability Assessment for Drinking Purposes Based on Water Quality Index. Int. J. Environ. Res. Public Health (2022) 19, 9071. https://doi.org/10.3390/ijerph19159071.
Wang T, Xu S, Liu J. Dynamic Assessment of Comprehensive Water Quality Considering the Release of Sediment Pollution. Water (2017) 9(4), 275. doi:10.3390/w9040275.
- Question: Methods:
- A diagram of the types of WCW tested should be included for a better presentation of the research.
- Information is needed on the number of water samples taken from each rainwater harvesting variant.
(Response)
Thanks for your advice.
We increased the number of samples in the Table 1.
As can be seen from Table 1, there are a total of 11 water cellar monitoring points (after excluding some overlapping samples), and each monitoring point is replicated three times, so there are 33 samples in total.
- Question: Results:
- Figures 3 and 4 should be described in more detail in the caption. Each drawing should be a complete reference object so that the reader does not have to search for explanations in the text (which is tedious). The symbols denoting each component of the figures (a) - h) in Fig. 3 and a) - d) in Fig. 4) should be explained, and symbols a-d should be replaced by others to avoid confusion, and explained under both figures.
(Response)
Thanks for your advice. We added instructions. W changed Fig.4 to Fig.4 and Fig.5.
Fig. 3 Statistical description of quality index of WCW at different catchment surfaces (a), at different depths (b), at different time (c), at different water types (d) in northwest Shanxi Province.
Fig. 4 Assessment of PCA at different catchment surfaces (1), at different depths (2), at different time (3), at different water types (4) in northwest Shanxi Province
Fig. 5 Assessment of NPI (1) and KI (2) of WCW at different catchment surfaces (A, B, C, D), at different depths (S1, S2, S3, S4), at different time (T1, T2, T3), at different water types (CW, RW, GW, TW) in northwest Shanxi Province (The letters such as a, b, c, d in the figure indicate the significant difference (Duncan) test at the P=0.05 level. Different lowercase letters (a–c) above the bars indicate significant differences (Duncan’s test, P < 0.05) among the treatments).
- Question: Discussion:
- The discussion should be conducted with the results of similar research papers from the world literature. The scope of works cited should be significantly expanded to include results from other countries and continents.
- The work presents a comparison of the determined values of the chemical indicators of water with the standards for drinking water. These standards also include microbiological indicators, which are essential for drinking water. Have such studies been done on water at WCW? Include a comment on this.
(Response)
-Thanks for your advice. We cite expert research from many more countries and continents. As shown in the L459-463, L469, L471-473, L479-480, L495.

Reviewer 3 Report
The manuscript ID # sustainability-1856929, MDPI, entitled “Evaluation of water quality of collected rainwater in northeastern Loess Plateau”, in which The quality of water cellars water of different rainwater catchments, different depths, and different storage times was evaluated by principal component analysis (PCA) and Nemerov’s pollution index (NPI) for northwest Shanxi province, China. The manuscript seems bit well prepared, and need some improvements. I suggest a major revision including enriching text (sentence and reference corrections), Figures captions revision and revision of graphical abstract.
Some of my specific comments are given below
GRAPHICAL ABSTRACT
1. The graphical abstract is too wordy, a pictorial graphical abstract would be more appropriate showing different heights of Water cellar along with their water quality.
MANUSCRIPT TEXT:
2. Line 66, “As an unconventional water resource, water cellars water (WCW) mainly converts …for the production and living use of residents”. Production of what? Sentence needs revision, similarly at line 79.
3. Line 81(Ministry of Housing and Urban-Rural Development of the People’s Republic of China), Line 84, (Ministry of Transport of the 84 People’s Republic of China), and, Line 87, (Central Government of the People’s Republic of China), Author needs to provide reference for these. Or any other reference verifying data from line 81 to 87.
4. Line 138, “WCW in the reservoir were collected rainwater that year and was fully stored at the end of the rainy season” sentence needs revision.
5. Line 216-218, “From Figure 1a, it can be seen that the annual precipitation in this area from 1990 to 2019 was between 350 and 500 mm, with an average value of 444.35 mm; the evaporation was between 1000 and 1800 mm, with an average value of 1374.85 mm”. First, it is not figure 1a but figure 2a, second the precipitation graph shows the data 350-600, not 300-500. And furthermore, evaporation graph shows between 800-1900. Author needs to clarify and match the graphical data with sentences.
6. Line 240-242, “The average content of F-, Cl-, SO42-, NO3-, Na+, NH4+, K+, Mg2+and Ca2+ was 0.07 mg/L, 2.84 mg/L, 2.00 mg/L, 0.11 mg/L, 0.71 mg/L, 0.16 mg/L, 0.17 mg/L, and 0.50 mg/L, respectively.” The values in Table 3 are not matching with the sentences and moreover of F- value is 0.007, not 0.7. Author needs to check these values and write correctly according to the order of names.
7. Table 3, Drinking water standards, are these World Health Organization (WHO) drinking water quality standards or The Drinking Water Standards of China, Author needs to clarify in manuscript earlier and also provide a reference.
8. Line 345, “The WCW (a total of 24 samples), rainwater, groundwater, and tap water in the survey site were compared (Fig. 3g, 3h). The ion concentration from high to low was UW > DW > RW > TW.”
9. Author needs to write full forms before using abbreviations. If Rainwater is (RW), Tap water is (TW) then ground water should be (GW), Author needs to clarify in text and also in the respective graphs.
10. Line 359, “It can be seen from Figure 3d that the WCW PI was 0.75…” It is Figure 4d.
GRAPHS
11. Figure 1. Map of study site, the caption should be more expressive like; Map of study site (North west Shanxi province, China).
12. Figure 3 and 4, The captions need explanation about each sub part of figure, for example Figure 4. Assessment of PCA and NPI of WCW at different depths (b), at different time (c)…..in northwest Shanxi Province.
13. Figure 3 and 4, Different lowercase letters (a–d) above the bars indicate significant differences (Duncan’s test, P < 0.05) among the treatments, but the author needs to explain somewhere that how much difference is there from 1 to d or e.
Author Response
Dear Reviewer 3:
Thank you very much for your suggestions and corrections. We also thanks for giving us a chance to improve our manuscript. All modifications are marked by yellow in revised manuscript, and we commissioned experts in native language to correct the English expressions.
GRAPHICAL ABSTRACT
- The graphical abstract is too wordy, a pictorial graphical abstract would be more appropriate showing different heights of Water cellar along with their water quality.
(Response)
Thanks for your advice. We revised the graphical abstract.
MANUSCRIPT TEXT:
- Line 66, “As an unconventional water resource, water cellars water (WCW) mainly converts …for the production and living use of residents”. Production of what? Sentence needs revision, similarly at line 79.
(Response)
Thanks for your correction. We revised the sentence. As shown in the L77-79, L89-90.
-As an unconventional water resource, water cellar water (WCW) mainly converts average atmospheric precipitation into the surface runoff, which then flows into water cellars for irrigation and drinking by residents.
-Therefore, the current status of WCW quality in rural areas is of great importance for the safety of drinking water and irrigation for rural residents.
- Line 81(Ministry of Housing and Urban-Rural Development of the People’s Republic of China), Line 84, (Ministry of Transport of the 84 People’s Republic of China), and, Line 87, (Central Government of the People’s Republic of China), Author needs to provide reference for these. Or any other reference verifying data from line 81 to 87.
(Response)
Thanks for your advice. We added the URL of the data source. As shown in the L93, L96-97, L100.
https://www.mohurd.gov.cn/xinwen/gzdt/202003/20200330_244697.html.
https://www.mot.gov.cn/jiaotongyaowen/202010/t20201023_3479335.html.
http://www.gov.cn/shuju/hgjjyxqk/xiangqing/np.html.
- Line 138,“WCW in the reservoir were collected rainwater that year and was fully stored at the end of the rainy season” sentence needs revision.
(Response)
Thanks for you correction. We revised it. As shown in the L171-172.
The water stored in the cellar is current rainwater and is completely stored at the end of the rainy season.
- Line 216-218, “From Figure 1a, it can be seen that the annual precipitation in this area from 1990 to 2019 was between 350 and 500 mm, with an average value of 444.35 mm; the evaporation was between 1000 and 1800 mm, with an average value of 1374.85 mm”. First, it is not figure 1a but figure 2a, second the precipitation graph shows the data 350-600, not 300-500. And furthermore, evaporation graph shows between 800-1900. Author needs to clarify and match the graphical data with sentences.
(Response)
Thank you very much for your correction. We revised the chart and data issues, and carefully checked the follow-up data. As shown in the L248, L249, L250.
As can be seen from Figure 2a, the annual precipitation in the region from 1990-2019 ranged from 300-600 mm with an average value of 444.35 mm, and the evaporation ranged from 800-1910 mm with an average value of 1374.85 mm.
- Line 240-242, “The average content of F-, Cl-, SO42-, NO3-, Na+, NH4+, K+, Mg2+and Ca2+ was 0.07 mg/L, 2.84 mg/L, 2.00 mg/L, 0.11 mg/L, 0.71 mg/L, 0.16 mg/L, 0.17 mg/L, and 0.50 mg/L, respectively.” The values in Table 3 are not matching with the sentences and moreover of F- value is 0.007, not 0.7. Author needs to check these values and write correctly according to the order of names.
(Response)
Thank you again for your corrections. As shown in the L269.
The total average concentrations of F-, Cl-, SO42-, NO3-, Na+, NH4+, K+, Mg2+and Ca2+ were 0.007mg/L, 0.07 mg/L, 2.84 mg/L, 2.00 mg/L, 0.11 mg/L, 0.71 mg/L, 0.16 mg/L, 0.17 mg/L, and 0.50 mg/L, respectively.
- Table 3, Drinking water standards, are these World Health Organization (WHO) drinking water quality standards or The Drinking Water Standards of China, Author needs to clarify in manuscript earlier and also provide a reference.
(Response)
Thanks for your advice. We added drinking water standards and added the national standard serial number in the manuscript.
Chinese drinking water standards GB 5749—2022. As shown in the Table 3.
- Line 345, “The WCW (a total of 24 samples), rainwater, groundwater, and tap water in the survey site were compared (Fig. 3g, 3h). The ion concentration from high to low was UW > DW > RW > TW.”
- Author needs to write full forms before using abbreviations. If Rainwater is (RW), Tap water is (TW) then ground water should be (GW), Author needs to clarify in text and also in the respective graphs.
(Response)
We revised all abbreviations in the manuscript, i.e. rainwater is (RW), tap water is (TW), ground water is (GW), cellars water is (CW). As shown in the L394, L413, L505.
The total ion concentrations from highest to lowest were GW (groundwater) > CW (cellar water) > RW (rainwater) > TW (tap water).
- Line 359,“It can be seen from Figure 3d that the WCW PI was 0.75…” It is Figure 4d.
(Response)
We revised it. Thanks for your correction. As shown in the L406.
GRAPHS
- Figure 1. Map of study site, the caption should be more expressive like; Map of study site (North west Shanxi province, China).
(Response)
Thanks for your suggestion. We revised the caption.
- Figure 3 and 4, The captions need explanation about each sub part of figure, for exampleFigure 4.Assessment of PCA and NPI of WCW at different depths (b), at different time (c)…..in northwest Shanxi Province.
(Response)
Thanks for your advice. We changed Fig.4 to new Fig.4 and Fig.5.
Fig. 3 Statistical description of quality index of WCW at different catchment surfaces (1-2), at different depths (3-4), at different time (5-6), at different water types (7-8) in northwest Shanxi Province.
Fig. 4 Assessment of PCA at different catchment surfaces (1), at different depths (2), at different time (3), at different water types (4) in northwest Shanxi Province
Fig. 5 Assessment of NPI (1) and KI (2) of WCW at different catchment surfaces (A, B, C, D), at different depths (S1, S2, S3, S4), at different time (T1, T2, T3), at different water types (CW, RW, GW, TW) in northwest Shanxi Province (The letters such as a, b, c, d in the figure indicate the significant difference (Duncan) test at the P=0.05 level. Different lowercase letters (a-c) above the bars indicate significant differences (Duncan’s test, P < 0.05) among the treatments).
- Figure 3 and 4, Different lowercase letters (a–d) above the bars indicate significant differences (Duncan’s test, P < 0.05) among the treatments, but the author needs to explain somewhere that how much difference is there from 1 to d or e.
(Response)
Thanks for your advice. We added instructions.
The letters such as a, b, c, d in the figure indicate the significant difference (Duncan) test at the P=0.05 level. Different lowercase letters (a–d) above the bars indicate significant differences (Duncan’s test, P < 0.05) among the treatments).

Reviewer 4 Report
Please change the graphical abstract to an appropriate form. In the present form it can't be considered a graphical abstract, instead it resembles a short textual abstract.
The overall impression is that the conducted research is interesting but poorly presented. I would advise the Authors to carry out a thorough review and try to complement the Manuscript with more suitable and detailed explanations to the implemented work and include more results. For example, the results of PCA could be presented in other more depicting ways (monoplot and other graphical representations) that would help better understand the attained conclusions. Considering the restricted amount of presented results, and explained study, the paper leaves the impression that it isn't finished yet.
Some exact issues are listed below:
Lines 138 and 139, sentence "WCW in the reservoir were collected rainwater that year and was fully stored at the end of the rainy season" is unclear. Could the Authors please clarify what they intended to state.
I think there is a typo at line 146 where it should probably say "cellar" instead of "caller".
Line 216 references Figure 1a, yet there is no such Fig. in the paper.
Figures 3. should be complemented with proper labels to allow the reader to differentiate them.
Lines 368 to 370 state "It can be seen from the quality evaluation results of different types of water that the results obtained by the PCA and the NPI methods were consistent.", could the Authors please provide adequate explanation t this conclusion.
Lines 495 to 498, sentence "The above conclusions further proved that the WCW in the rural area of the northeastern Loess Plateau was clean, and could be consumed by humans and animals after scientific management." should be altered and backed up with reference, or excluded.
Author Response
Dear Reviewer 4:
First of all, we are grateful to you for approval our manuscript, and we also thanks for your suggestions for improving the manuscript. All modifications are marked by yellow in revised manuscript, and we commissioned experts in native language to correct the English expressions.
- Question: Please change the graphical abstract to an appropriate form. In the present form it can't be considered a graphical abstract, instead it resembles a short textual abstract.
(Response)
Thanks for your advice. We revised the graphical abstract.
- Question: The overall impression is that the conducted research is interesting but poorly presented. I would advise the Authors to carry out a thorough review and try to complement the Manuscript with more suitable and detailed explanations to the implemented work and include more results. For example, the results of PCA could be presented in other more depicting ways (monoplot and other graphical representations) that would help better understand the attained conclusions. Considering the restricted amount of presented results, and explained study, the paper leaves the impression that it isn't finished yet.
(Response)
Thank you for acknowledging the experiment, and we are very sorry for the immaturity of the writing and the lack of language expression. We checked the manuscript, and in Section 3.4, the correlation analysis of depth and ions was added. We have made detailed changes in other parts
We revised the Fig.4. As shown in Fig.4 and Fig.5.
- Question: Lines 138 and 139, sentence "WCW in the reservoir were collected rainwater that year and was fully stored at the end of the rainy season" is unclear. Could the Authors please clarify what they intended to state.
(Response)
Thanks for you correction. We revised it. As shown in the L171-172.
The water stored in the cellar is current rainwater and is completely stored at the end of the rainy season.
- Question: I think there is a typo at line 146 where it should probably say "cellar" instead of "caller".
(Response)
Very sorry for our mistake, we corrected the word. As shown in the L179.
- Question: Line 216 references Figure 1a, yet there is no such Fig. in the paper.
(Response)
This is our mistake, here was Fig.2a, we corrected it and checked the subsequent figures. As shown in the L248.
- Question: Figures 3. should be complemented with proper labels to allow the reader to differentiate them.
(Response)
Fig. 3 Statistical description of quality index of WCW at different catchment surfaces (1-2), at different depths (3-4), at different time (5-6), at different water types (7-8) in northwest Shanxi Province (The letters such as a, b, c, d in the figure indicate the significant difference (Duncan) test at the P=0.05 level. Different lowercase letters (a–d) above the bars indicate significant differences (Duncan’s test, P < 0.05) among the treatments)
- Question: Lines 368 to 370 state "It can be seen from the quality evaluation results of different types of water that the results obtained by the PCA and the NPI methods were consistent.", could the Authors please provide adequate explanation t this conclusion.
(Response)
According to the Nemerow index calculation, rainwater, tap water, and cellar water are all grade I water (clean water), and groundwater is grade III water (moderately polluted water). According to the PCA, the quality of different types of water was sorted in descending order as rainwater > tap water > cellar water > groundwater. In summary, rainwater (I) > tap water (I) > cellar water (I) > groundwater (III).
It can be seen from the quality evaluation results of different types of water that the results obtained by the PCA and the NPI methods were consistent and complement each other.
Liu M, Chen SJ. Groundwater quality assessment of Honghu Area based on the Nemerow Index and principal component analysis method. Journal of Central China Normal University(Natural Sciences), (2016) 50(04): 633-640. DOI:10.19603/j.cnki.1000-1190.2016.04.025 (In Chinese with English abstract).
As shown in the L413-418.
- Question: Lines 495 to 498, sentence "The above conclusions further proved that the WCW in the rural area of the northeastern Loess Plateau was clean, and could be consumed by humans and animals after scientific management." should be altered and backed up with reference, or excluded.
(Response)
Thanks for your suggestion, we removed this sentence.

Round 2
Reviewer 2 Report
I accept the revised version of the manuscript. It can be considered for publication in Sustainability.
Reviewer 3 Report
The manuscript is much improved and I am satisfied with the comments of author and accept this manuscript for publication.
NOTE: The references at line 85, 88, 92 are only URLS which should be corrected according to journal format by using endnote or mendeley software in preprint before publishing final version.
Reviewer 4 Report
The Manuscript is missing the Figures. Could the Authors please include them and submit the Article again.